# Evidence of Phase Transitions and Their Role in the Transient Behavior of Mechanical Properties and Low Temperature Degradation of 3Y-TZP Made from Stabilizer-Coated Powder

**Frank Kern**

Institute for Manufacturing Technologies of Ceramic Components and Composites (IFKB), University of Stuttgart, Allmandring 7b, D-70569 Stuttgart, Germany; frank.kern@ifkb.uni-stuttgart.de; Tel.: +49-711-685-68301

**Abstract:** The substance 3 mol% yttria stabilized zirconia (3Y-TZP) has become a commodity for the manufacture of components in biomedical and engineering applications. Materials made from stabilizer-coated rather than co-precipitated starting powders are known for their superior toughness and low temperature ageing resistance. The reason for this phenomenon is however still not fully understood. In this study, 3Y-TZP materials hot pressed at 1300–1450 °C for 1 h were characterized. It was found that at a sintering temperature of 1375 °C, a transition from fine grain to coarse grain microstructure associated with a shift from tough and ageing resistant to brittle and prone to ageing was observed. The detailed analysis of the phase composition by X-ray diffraction revealed that TZPs consists of up to five crystallographically different phases of zirconia simultaneously whose contents dynamically change with sintering temperature. At low sintering temperature, the predominant phases are a tetragonal phase with low yttria content and large domain size and high tetragonality together with a cubic phase of high yttria content. At high temperature, a tetragonal phase of higher yttria content and lower tetragonality is formed together with a cubic phase of lower yttria content.

**Keywords:** zirconia; microstructure; mechanical properties; phase composition; low temperature degradation

## 1. Introduction

The excellent mechanical properties of partially stabilized zirconia materials is based on an effect called "transformation toughening" which describes the stress induced martensitic transformation of metastable tetragonal phase associated with volume expansion and shear. Along a proceeding crack, a process zone is formed which exerts compressive force on the crack and reduces the stress intensity at the crack tip [1,2].

In order to retain the high temperature tetragonal phase at room temperature and to prevent spontaneous transformation during cooling, stabilizer oxides such as magnesia, calcia, yttria or ceria are added. Today yttria is the most frequently used stabilizer oxide for applications in dental restoration requiring high strength and moderate toughness [3]. Aliovalent yttria replaces zirconia in the lattice and leads simultaneously to the formation of oxygen vacancies which change the coordination of zirconia and stabilize the tetragonal phase [4]. With respect to thermodynamics, a stabilizer content close to 3 mol% $Y_2O_3$ is used, this composition is located in the t + c field (a miscibility gap) [5]. At the typical sintering temperatures of 1400–1500 °C, ~80% tetragonal phase coexist with ~20% cubic phase if equilibrium conditions are reached.

Beside the excellent mechanical properties there is, however, one major drawback which is low temperature degradation (LTD) [6–9]. LTD is caused by the diffusion of the hydroxyl-group

containing molecules such as water into the oxygen vacancies. Tetragonal zirconia forced into a sevenfold coordination by stabilization with yttria then returns into the unfavorable eight-fold coordination (presenting a size mismatch at ambient temperature) and transforms spontaneously to monoclinic; an effect which can cause severe difficulties in biomedical applications by spallation or even complete disintegration of implants in vivo [10,11]. This effect is thermally activated but also happens at ambient temperature with measurable speed.

Nowadays yttria stabilized zirconia (Y-TZP) powders are produced by co-precipitation of yttria and zirconia from inorganic precursors, this leads to a homogeneous distribution of yttria at the atomic level [12]. The substance, 3 mol% yttria stabilized zirconia (3Y-TZP) from such commercially available co-precipitated powders form tetragonal phase only—if sintered at low temperature. Despite supersaturation with yttria, cubic phase is only formed at high sintering temperature and long dwell [13,14]. Yttria segregates to the grain boundaries so that boundaries contain significantly more yttria than the bulk of the grains [15]. This solute drag effect is also responsible for the extremely fine grain size of Y-TZP. Phase transformation in Y-TZP is locally limited, transformation zones are comparatively small and the R-curves are thus very steep [1]. Due to crystallographic limitations and lower symmetry compared to Ce-TZP the growth of martensite plates into neighbor grains and the triggering of consecutive transformation of domains with similar orientation are hindered. Thus the phase transformation in Y-TZP is predominantly dilatoric [2].

Y-TZP materials made by a "coating" processes with excellent fracture and low temperature degradation resistance were reported in the late 1990s by Singh and Burger [16,17]. The starting powder for these investigations provided by an industrial supplier was however not actually stabilizer "coated". Plasma atomized zirconia nanopowder was intensely co-milled with yttria to achieve a very homogeneous stabilizer distribution. Ohnishi also studied Y-TZPs made by intensive co-milling over a wide range of yttria contents and sintering temperatures [18,19]. The "coating" concept was later revitalized by the group of Van der Biest who used yttrium nitrate coated onto the monoclinic starting powder as yttria precursor [20,21]. Some studies by the authors using a slightly modified coating process confirmed the early findings by Singh, Ohnishi and Burger. In a recent publication by Zhang stabilizer concentration gradients in the regions of the grain boundaries were measured and found to be more pronounced in case of "coated" compared to "co-precipitated" material [22].

The current scientific consensus is the following: in TZP made from co-precipitated powder, the bulk of the grains is initially supersaturated with yttria. The thermodynamic driving force to segregate yttria leads to an yttria enrichment of the grain boundary phase and finally to the formation of cubic grains having high and tetragonal grains having low yttria content. The segregation is kinetically hindered as yttria has to diffuse "up-hill" against the concentration gradient. In case of "coated" materials the stabilizer is initially located entirely outside the monoclinic grains and is incorporated by diffusion during sintering, Burger showed that the process starts slightly above the eutectoid temperature of ~1150 °C [17]. The stabilizer thus initially forms a gradient. It is believed that the grains initially have an yttria-rich and non-transformable (by either stress or LTD) shell and a low yttria content core which is very transformable (and tough). Early separation of cubic phase is believed to be favored as the stabilizer does not need to diffuse against the concentration gradient [22,23].

Typically some monoclinic phase is found by X-ray diffraction (XRD), it can however not be detected by scanning electron microscopy (SEM). The favorable mechanical and LTD properties vanish at high sintering temperature and the properties of "coated" and "co-precipitated" TZP become very similar.

The current study aims at elucidation of what actually happens in detail at which temperature and in which sequence. For this purpose materials were made by hot pressing at relatively low sintering temperatures and short dwell to obtain materials that are fully dense (thus different elastic constraint cannot effect phase composition) but also to be able to monitor the stabilizer distribution in the early stage by XRD. A complete characterization of mechanical properties, microstructure and low

temperature degradation behavior is included in order to be able to correlate the phase changes to macroscopically measurable and technically relevant properties based on a uniform set of data.

## 2. Materials and Methods

The manufacturing of the stabilizer-coated powders by the nitrate route was first described by Yuan [20]. The modified procedure applied here has been shown by the authors in detail elsewhere [23,24]. The starting powder was a nanoscale monoclinic powder ($S_{BET}$ = 15–17 m$^2$/g). Yttria (purity 99.9%) was dissolved in five N $HNO_3$ and added to zirconia dispersed in 2-propanol and milled overnight, 0.5 vol.% alumina ($S_{BET}$ = 8 m$^3$/g) was added as a sintering aid and in order to hinder LTD. The dispersion was then dried, calcined in air in two stages at 350 °C and 600 °C with an intermittent grinding and finally attrition milled in 2-propanol. The dried powder was then pressed into cylindrical disks of 45 mm diameter by hot pressing in vacuum at 1300–1450 °C for 1h at 50 MPa axial pressure. Two samples were pressed at a time in a boron nitride clad graphite die.

Samples were subsequently lapped and polished with 15 μm, 6 μm and 1 μm diamond suspension until a mirror-like surface was achieved. The entire disks were used for measuring the density by buoyancy method and the Young's modulus by resonance frequency method (IMCE, Genk, Belgium). The polished disks were then cut into bending bars of 4 × 2 mm$^2$ diameter. The sides of the bars were lapped with 15 μm suspension and the edges were beveled with a 20 μm diamond disk to remove any preparation induced defects.

The mechanical characterization included the measurement of Vickers hardness HV10 (Bareiss, Oberdischingen, Germany) and the measurement of fracture resistance by direct crack length measurement (DCM) and indentation strength in bending (ISB) (Zwick, Ulm, Germany). For the DCM measurements the size of five HV10 indents and the cracks evolved at indentation corners were measured by optical microscopy. The fracture resistance was calculated by the model of Anstis [25] (this model was used in order to be able to compare toughness data with a previous publication by Zhang [22]). For the ISB tests, five bending bars were notched with a HV10 indent in the middle of the tensile side with the side cracks parallel and perpendicular to the sides. The residual strength was measured immediately after that in a 4 pt bending test setup with 20 mm outer and inner span at a crosshead speed of 2.5 mm/min to avoid subcritical crack growth (Zwick, Germany). The fracture resistance was calculated according to the model of Chantikul [26].

The microstructure was studied by SEM (Zeiss Gemini, Oberkochen, Germany, in lens SE mode, 3kV acceleration voltage). Polished samples were etched in hydrogen at 1200 °C for 5 min to reveal the grain boundaries (Xerion, Freiberg, Germany). Fracture faces were studied to obtain an impression of inter-or transgranular fracture of samples sintered at different temperatures. The grain sizes were determined by linear intercept method using a geometry factor of 1.56 [27].

The LTD tests were carried out by exposing Y-TZP samples to a saturated water vapor atmosphere in a stainless steel autoclave at 134 °C (vapor pressure 3 bar) for 1–100 h in a log10 time scale (1 h, 3 h, 10 h, 30 h and 100 h).

The phase composition of polished surfaces, fracture surfaces and autoclave aged surfaces was measured by XRD (X'Pert MPD, Malvern Panalytical, Royston, UK, Bragg Brentano setup, CuKα1). For determination of the monoclinic content the three characteristic peaks in the 26–33° 2θ-range were integrated and the monoclinic content was determined according to the calibration curve of Toraya [28]. Transformation zone sizes were calculated according to Kosmac [29] and the transformation toughness increments were calculated according to the formula of McMeeking and Evans [30]. As the (101) tetragonal and (111) cubic peaks at ~30° 2θ coincide and cannot be separated, the cubic and tetragonal contents were analyzed by integrating the tetragonal (004) and (400) peaks and the cubic (400) peak in the 72–76° 2θ-range [31].

## 3. Results

### 3.1. Microstructure

Figure 1a–c show the thermally etched surfaces of 3Y-TZP materials sintered at 1300 °C, 1375 °C and 1450 °C. The corresponding fracture surfaces are shown in Figure 1d–f. Grain sizes determined by linear intercept method are shown in Figure 2. SEM images confirm density measurements that materials always reach full density.

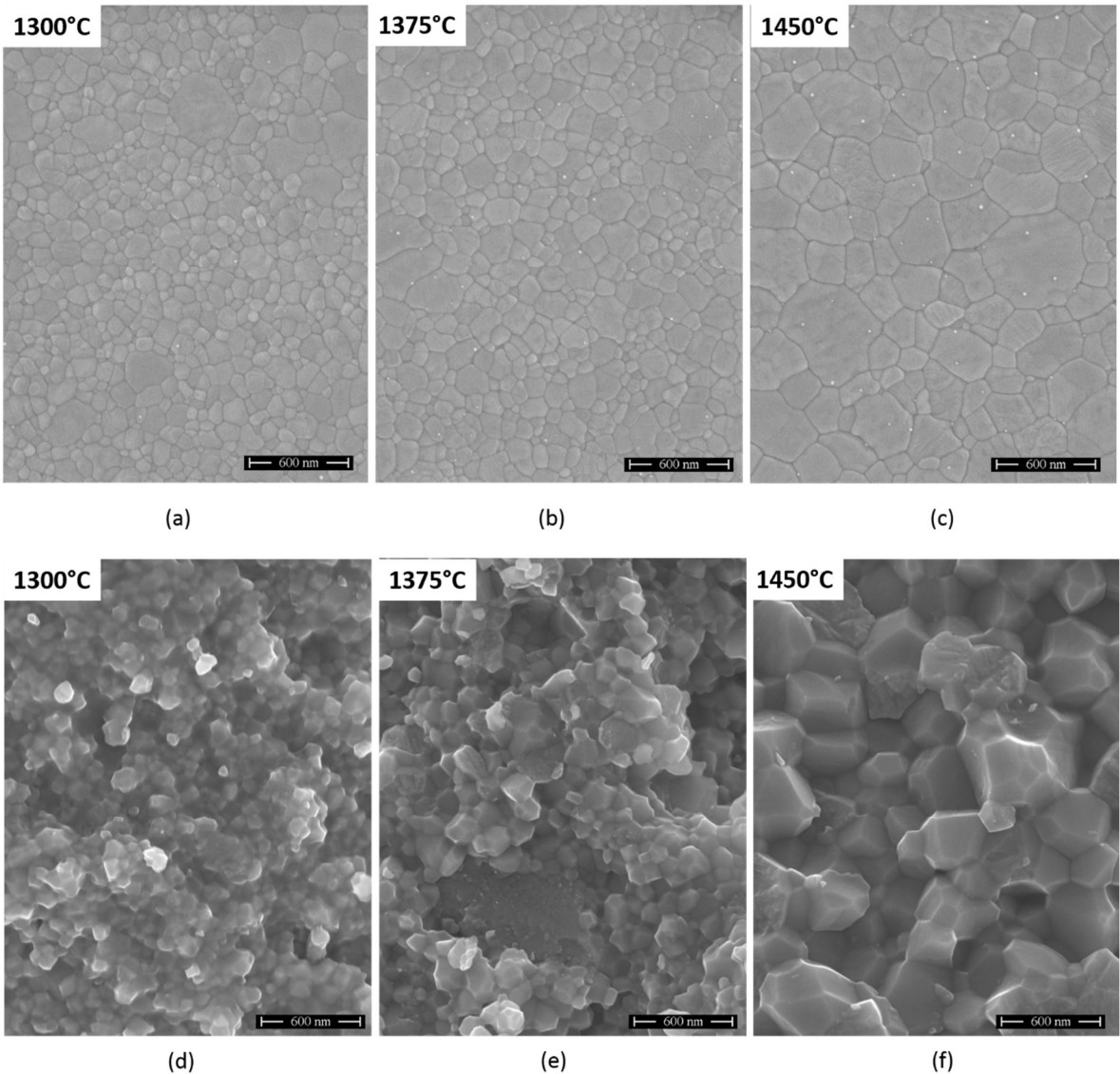

**Figure 1.** SEM images of thermally etched surfaces of 3 mol% yttria stabilized zirconia (3Y-TZP) hot-pressed at 1300 °C/1h (**a**), 1375 °C/1h (**b**) and 1450 °C/1h (**c**) and fracture surfaces of 3Y-TZP hot-pressed at 1300 °C/1h (**d**), 1375 °C/1h (**e**) and 1450 °C/1h (**f**).

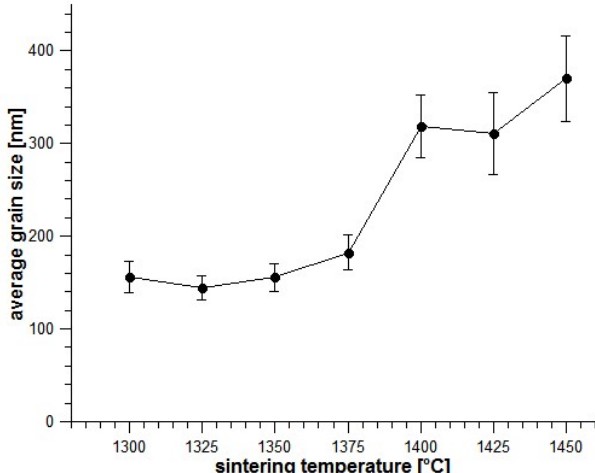

**Figure 2.** Grain sizes of 3Y-TZP hot pressed at 1300–1450 °C/h measured by the linear intercept method.

As can be seen from SEM-images the grain sizes grow with increasing temperature as expected. The grain growth, however, was discontinuous, between 1300 and 1350 °C the grain size stayed almost constant at ~ 150 ± 10 nm, then the average grain size grew to 190 nm at 1375 and to 300–350 nm at 1400–1450 °C. At low sintering temperatures a clearly bimodal grain size distribution is observed with small grains of ~100 nm size and mostly isolated larger grains of up to 500 nm. At temperatures > 1400 °C the small grains disappeared completely and a broad grain size distribution between 200–600 nm was obtained. The fracture surfaces changed drastically with sintering temperature. At 1300 °C quite diffuse fracture images can be observed, the crack mode of the small grains was purely intergranular and the individual grains were not well faceted. Transgranular fracture can only be observed for single large zirconia grains which are presumably cubic (Figure S13 in the Supplementary Materials). Towards 1375 °C the grains in the fracture surfaces become more faceted, fracture was still predominantly intergranular but the fraction of transgranular fracture distinctly increased to approximately 10–20%. At the highest sintering temperature, a mixed mode fracture was observed with—roughly estimated—about 30% of grains fractured transgranularly. Grains featuring transgranular fracture show a surface with a nanoscale roughness. Grains of the alumina dopant (Figure 1e, large grain at the bottom, Figures S10 and S13 in the Supplementary Materials) fracture transgranularly for all sintering temperatures. The supplement contains SEM images (magnification 20,000×) of thermally etched microstructures and fracture surfaces for all sintering temperatures.

### 3.2. Mechanical Properties

Figure 3 shows the hardness and Young's modulus of 3Y-TZP materials at different sintering temperatures. Irrespective of sintering conditions the Young's modulus stays more or less constant at the typical value of 210 GPa known from literature [1]. The Vickers hardness HV10 increased linearly from 1230 HV10 to 1330 HV10 between 1300 °C and 1400 °C sintering temperature then a constant value is observed. Fine grain size did not lead to a hardening effect as might be expected from the Hall–Petch equation [32].

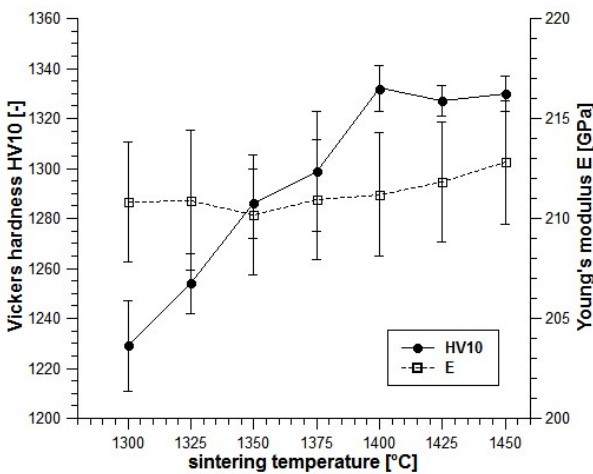

**Figure 3.** Vickers hardness HV10 and Young's modulus E of 3Y-TZP vs. sintering temperature.

Fracture resistance determined by the DCM and ISB methods are plotted in Figure 4. Independent of the measuring technique the toughness evolution shows a S-shape with a relatively sharp transition between high toughness of ~10 MPa$\sqrt{m}$ (ISB) at low sintering temperature and low toughness ~5 MPa$\sqrt{m}$ at high sintering temperature, the turning point can be located at 1375 °C. DCM data evaluated using the Anstis model were always lower, the gap between Anstis and ISB data narrowed with increasing sintering temperature from 2 MPa$\sqrt{m}$ at 1300–1350 °C to 1 MPa$\sqrt{m}$ at 1400–1450 °C.

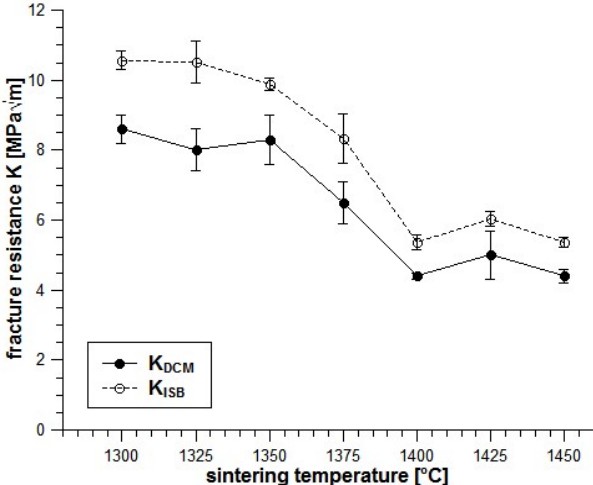

**Figure 4.** Fracture resistance of 3Y-TZP measured by the direct crack length measurement (DCM) and indentation strength in bending (ISB) method vs. sintering temperature.

ISB-based toughness values were in the same range as recently reported for 3Y-TZP and 3Yb-TZP. The DCM values for the high sintering temperature range were in good accord with data published recently by Zhang [22].

*3.3. Phase Composition*

The monoclinic contents in polished surfaces $V_{m,pol}$ representing the bulk composition and fracture surfaces $V_{m,F}$ are shown in Figure 5. The resulting difference $V_f = V_{m,F} - V_{m,pol}$ represents the transformed fraction during fracture ("transformability").

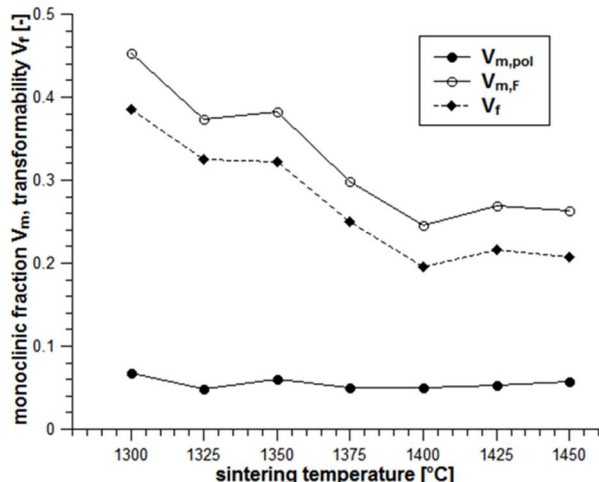

**Figure 5.** Monoclinic contents $V_m$ in polished and fractured surface and transformability $V_f$ of 3Y-TZP vs. sintering temperature.

The materials always contain a certain amount of monoclinic, the variation of $V_{m,pol}$ with sintering temperature is negligible and in the range of the measuring accuracy of $\pm1$ vol%, all samples contained 5–6 vol.% monoclinic. Contrary to this, the monoclinic fraction in the fracture surfaces showed a clear trend to decline with increasing sintering temperature. Similar to the trend in fracture resistance this trend was discontinuous, transformability was high (33–38%) at low sintering temperature. It declinds sharply between 1350 °C and 1400 °C and levelled off at 20% between 1400 °C–1450 °C. XRD measurements carried out to quantify the amount of cubic and tetragonal phase in the 2θ-range between 72–76° 2θ delivered new and completely unexpected results.

As can be seen in Figure 6 there were not only the (004) and (400) peaks of tetragonal and the (400) peak of cubic phase; there were—depending on sintering temperature—up to six peaks simultaneously. At low sintering temperature the dominant peaks were $(004)_{t1}$ and $(400)_{t1}$ peaks located at 72.8° and 74.6° 2θ. A cubic peak $(400)_{c1}$ was located at 73.7° 2θ. At high sintering temperature of 1450 °C, the dominant peaks were $(004)_{t2}$ and $(400)_{t2}$ peaks located at 73.05° and 74.4° 2θ. A cubic peak $(400)_{c2}$ was located at 73.9° 2θ. In the transition range between high and low temperature such as at 1375 °C all six peaks appear simultaneously. While the phases indexed with a "1" disappear with increasing temperature the phases indexed with "2"start to evolve.

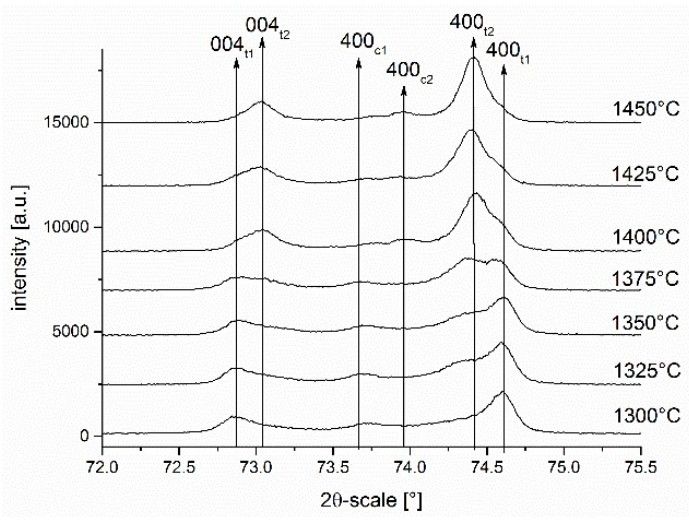

**Figure 6.** XRD traces of 3Y-TZP sintered at 1300–1450 °C in the 72-76° 2θ-range.

The calculation of the c/a ratio of the tetragonal phases (Figure 7) and the calculation of the cell parameter c of the cubic phases can provide further evidence. The t1 phase appearing at low sintering temperature had a high c/a ratio = tetragonality of ~1.021. The t2 phase had a significantly lower tetragonality of ~1.015. Based on experimental data by Lefevre and Scott [33,34] a tetragonality of 1.021 (and cell parameter of c = 519 pm) corresponds to an (extrapolated) yttria content in the tetragonal phase of 1–1.5 mol% $Y_2O_3$. A tetragonality of 1.015–1.016 (c = 517.7 pm) corresponds to an yttria content of 2.5 mol% $Y_2O_3$ which is the composition at the t/t + c phase boundary.

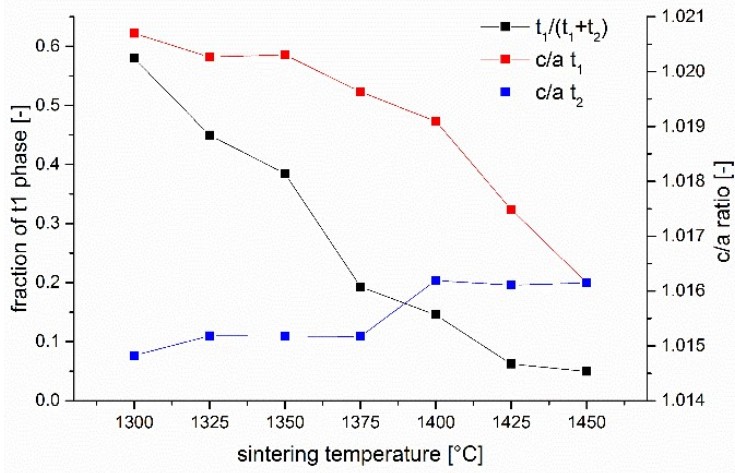

**Figure 7.** The c/a ratio (tetragonality) and estimated relative content of the t1 and t2 phase vs. sintering temperature.

By using a peak separation software, it was possible to separate and integrate the peaks and estimate the relative content of the tetragonal phases t1 and t2 at different temperatures. It can be seen that the fraction of the high tetragonality phase shows a linear decline with increasing sintering temperature.

An analysis of the width of t1 and t2 peaks shows that the t1 peaks were narrower and that domains of the t1 phase were therefore initially larger than the domains of the successively formed t2 phase. Cubic peaks were relatively broad initially indicating a small domain size.

The calibration curve of Scott [34] also enables calculating the yttria content of the two cubic phases. Accordingly, the c2 phase appearing at 73.7° 2θ has a cell parameter c = 513.5 pm and thus an yttrium content of 4–4.7 mol% $Y_2O_3$, the c1 phase appearing at 73.9° 2θ has a cell parameter c = 512 pm corresponding to a yttrium content of 7.5–8.3 mol% $Y_2O_3$. As peaks are broad, the calculated stabilizer contents should be understood as average values.

We may conclude from these XRD measurements that at low sintering temperature, a highly tetragonal phase of low yttria content coexists with a cubic phase of high yttria content and that an increase of the sintering temperature leads to a subsequent formation of a tetragonal phase with much lower tetragonality and a cubic phase which contains much less stabilizer. The transition between these two states is continuous.

### 3.4. Low Temperature Degradation

Figure 8 shows the monoclinic contents in 3Y-TZP materials sintered at 1300–1450 °C after an accelerated autoclave ageing test carried out at 134 °C for 3–100 h. Evidently the monoclinic contents after 3h ageing for materials sintered between 1300–1375 °C are identical to the unaged materials (Figure 5) within the error margins. Materials sintered at 1400–1450 °C contain 8–10 vol.% monoclinic (2–4 vol.% formed by LTD). At 30 h ageing time, the materials sintered between 1300–1350 °C still showed no significant increase in monoclinic content while the material sintered at 1375 °C has already contains 18 vol.% of monoclinic. The materials sintered at higher temperature were almost entirely

transformed. The cubic fraction will not transform so that the maximum transformable fraction $V_{m,max}$ should be in the range of 0.8, which is in line with measured monoclinic contents. At 100 h ageing the materials sintered at the two lowest temperatures are slightly affected by LTD (12–14 vol.% monoclinic). A surprising fact is that the material sintered at 1400 °C (80 vol.% at 30 h) seems more prone to ageing than the materials sintered at higher temperature (66–74 vol% after 30 h).

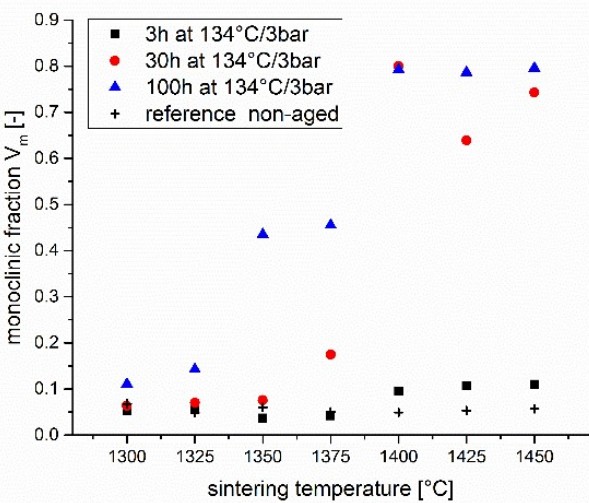

**Figure 8.** Low temperatured degradation behavior, monoclinic contents of 3Y-TZP sintered at different temperatures after accelerated autoclave ageing test at 134 °C/3bar water vapor pressure.

## 4. Discussion

The inverse grain size toughness correlation in Y-TZP ceramics made from monoclinic starting powders either very homogeneously mixed or coated with the stabilizer oxide was shown in many previous publications by different authors [16,17,22–24]. The experimental evidence however stands in sharp contrast to proven findings on Y-TZP materials made from co-precipitated powders which show a positive grain size - toughness correlation. They represent the state-of-the-art which makes the new results on coated powders hard to accept.

Distinct differences in grain boundary chemistry between co-precipitated and coated powders show that materials are substantially different. This was known before from transmission electron microscopy (TEM) investigations carried out by different authors [19,22]. TEM data are however only available for materials sintered at relatively high temperature. For materials sintered at extremely low temperature having grain sizes of ~150 nm a lateral resolution of <10 nm coupled with low scattering would be required which brings even modern STEM technology to its limits. Moreover concentration profiles determined by TEM cannot be directly correlated to phase compositions. XRD using an accelerator detector guaranteeing high signal quality at moderate measuring times of ~2 h even at high 2θ-range and low peak intensities was able to record the evidence of phase changes which were previously unknown. Such XRD investigations had already been carried out by Ohnishi, a quantitative interpretation was not given probably due to a lack of resolution [18]. In the present study the phase transitions encountered during sintering of "coated" 3Y-TZP have been shown quantitatively for the first time and they provide the missing facts to fully understand the processes taking place.

In order to visualize the diffusion and grain growth processes taking place with increasing sintering temperature a schematic drawing is shown in Figure 9.

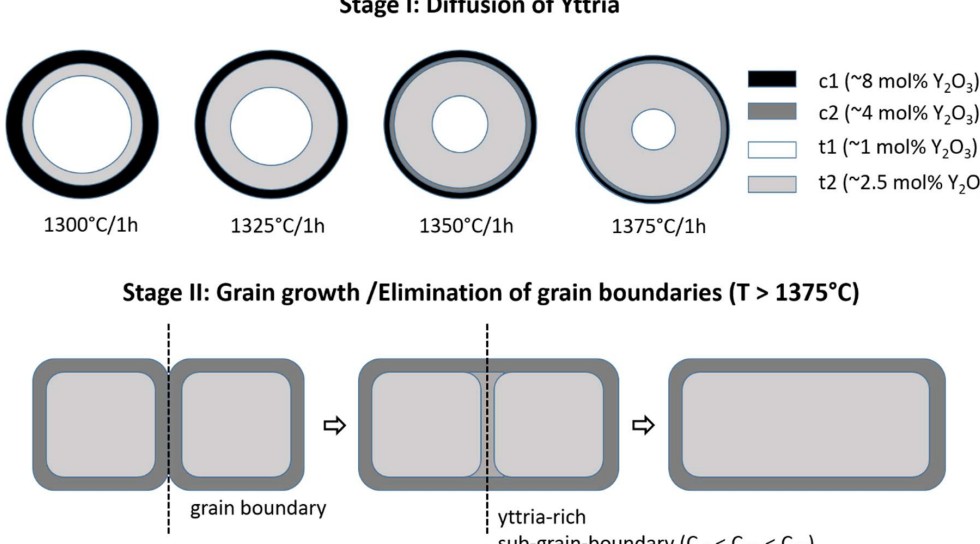

**Figure 9.** Yttria diffusion and grain growth, schematic diagram.

The stabilizer oxide initially located outside the zirconia grains progressively diffuses into the zirconia grains. It is trivial that the stabilizer concentration must decline from grain boundary to the center of the grain. What is new is the fact that the process happens stepwise and that the crystallographic phases formed can be tracked and described. In the first stage of the diffusion process (1300 °C/1 h) a thin yttria-rich c1 (8–9 mol% $Y_2O_3$) layer is formed at the grain boundary, while the bulk of the grain contains just a small amount of stabilizer (~1–1.5 mol% $Y_2O_3$). This t1 tetragonal phase forms large domains. The fact that the outer flanks of the $(004)_{t1}$ and $(400)_{t1}$ peaks are steep confirms this assumption. T1 has a high tetragonality and presumably a very high transformability. Between the high yttria cubic layer c1 at the boundary and the low yttria tetragonal t1 phase in the core a thin layer (broader peaks) of "regular" tetragonal t2 phase containing ~2.5 mol% $Y_2O_3$ is formed even at the lowest temperature.

With increasing sintering temperature the trend to an equilibration of yttria concentration proceeds (stage I in Figure 9). The tetragonal domains t1 in the center of the grains shrink and finally vanish at 1425 °C, the high yttria cubic phase c1 which acts as the source of yttria, disappears at the same temperature. What is left at temperatures > 1400 °C is a low yttria cubic phase c2 with 4–4.5 mol% $Y_2O_3$ and a tetragonal phase t2 which contains ~2.5 mol% $Y_2O_3$. This represents more or less the thermodynamic saturation level.

This interpretation can also explain the observations made in a recent publication [24] where 2.6–3Y-TZP materials had identical toughness at low sintering temperature but very different characteristics at higher sintering temperature.

The yttria concentration of the low yttria cubic phase c2 measured at the highest sintering temperature corresponds very well to the yttria contents measured by Zhang (4–6 mol%) by STEM for a sample sintered at 1550 °C [22].

The transition between these two extremes is reached at a sintering temperature of 1375 °C. Here coexistence of two tetragonal and two cubic phases is observed.

This transition temperature also marks the onset of grain growth and macroscopically detectable changes from tough to brittle and from highly LTD resistant to low LTD resistant. Moreover here the fracture mechanism changes from intergranular to transgranular and the fracture faces become more and more faceted.

Grain growth (stage II in Figure 9) is observed when two or more grains merge. The final structure of the new grain is not formed immediately, in the first stage of grain growth two grains coalesce, thereby the yttria rich shell at the junction is subducted and a yttria-rich sub-grain-boundary is formed

which contains almost as much yttria as the original grain boundary. As such high yttria concentrations are thermodynamically not tolerable the surplus of yttria subsequently diffuses out. Figure 8 shows that the materials are seemingly most vulnerable to ageing in this transition state where the cubic has not yet diffused out of the core completely and the untransformable cubic shell is thin and prone to hydrolysis. In Figure 1c (1450 °C) such former sub-grain boundaries are visible as lines with slightly darker shade in large grains.

The excellent ageing resistance of the materials sintered at low temperature is surprising taking into account that stress induced transformability is so high. In polished surfaces grains are sectioned so that probably the core of the grains which is almost unstabilized is subjected to the influence of the polar liquid. However most probably the first layer of grains (0–150 nm thickness) transforms immediately after polishing due to the release of the elastic constraint and is already monoclinic before ageing. The effect of transformation stresses induced by this initial process is low as stress caused by a spherical inclusion scales with $R^3$ and as the grain radius R is very small [35]. If we compare the residual stress caused by a transformed grain with R = 75 nm and a grain with R = 200 nm (which still fulfils the dental standard EN 6872 requiring a grain size D = 2R = 400 nm) the stress caused by the transformed smaller inclusion is just 5 % of the stress caused by the larger one. This stress is probably not high enough to trigger weakening of grain boundaries or even breakout of grains which would lead to subsequent penetration of the liquid into the bulk [9,11]. The results concerning LTD of 3-YTZ sintered at low temperature are comparable to a recent study carried out with a similar composition and sintering temperature (2 vol.% alumina, sintering temperature 1350 °C) [36].

The further growth of the transformation zone according to the accepted theory however requires hydrolyzing the yttria stabilizer present at the grain boundaries separating the first layer of grains from the next one [9,11]. As the grain boundary region of samples sintered at low temperature is extremely yttria-rich (~8 mol% $Y_2O_3$) and rather thick (>20 nm), a destabilizing of the grain boundary is not observed within the first 30 h of accelerated ageing at 134 °C (~100 years in vivo by rule of the thumb).

As with increasing time and temperature, the supersaturated cubic shell of each grain successively releases yttria into the bulk of the grain, which takes up yttria until the t/t + c phase boundary concentration is reached, its thickness and yttria concentration progressively sinks so that the ageing resistance declines (even though the stress induced transformability also declines). Once the c1 phase disappears the high ageing resistance is lost.

Another critical point to be addressed is the mechanistic interpretation of the transformation toughening processes. In co-precipitated or arc-melted and crushed Y-TZP the situation is quite clear: If a phase transformation is triggered by applied stress an initial martensite plate is formed, this martensite plate grows into the surrounding Y-TZP material and triggers the growth of new martensite domains. Due to its lower symmetry compared to Ce-TZP passing the martensite from one domain to the other is hindered so that Y-TZP forms much smaller transformation zone sizes but has a higher transformation efficiency than Ce-TZP. Still the transformation is predominantly dilatoric [2]. We may assume that this is also the case for the "coated" materials sintered at high temperature which are very similar to conventional Y-TZP. In such a situation self-accommodation of tetragonal domains and activating the shear component of transformation toughening is not favored [2].

In case of the materials sintered at low temperature this situation is probably different. The tetragonal grains contain only a small amount of stabilizer and are isolated from each other by a rather thick untransformable cubic layer. We may assume that the transformation behaviour therefore resembles the situation in Mg-PSZ or ZTA where zirconia inclusions are present in an untransformable matrix [37–39]. In this case triggering of neighbor zones cannot be achieved by growth of martensite plates from one domain to the other. The next domain has to be activated by applied stress. In such a situation self-accomodation of tetragonal domains is favored and the shear dependent part of the t-m phase transformation can be exploited. This was already supposed in an earlier publication by the author however only supported by an a priori assumption and a misfit of experimental data to the accepted theory [23]. The "real" transformation toughness increments

are however difficult to calculate. This starts with the calculation of the transformation zone size h according to Kosmac [29]. The transformation zone size h is calculated as:

$$h = (\sin\theta/2\mu)\cdot(V_{m,pol} - V_{trans})/(V_{m,F} - V_{trans}). \qquad (1)$$

While the X-ray absorption coefficient can be taken from literature and $V_{m,pol}$ and $V_{m,F}$ can be measured by XRD it is unclear which value to choose for $V_{trans}$, the transformable fraction of zirconia. For a homogeneous entirely tetragonal material $V_{trans} = 1$ but for sure we have to subtract the content of non-transformable cubic and possibly also the amount of non-transformable tetragonal. The t2 phase in material sintered at 1300 °C has a tetragonality of 1.0148 (c = 517.7 pm, yttria content ~3 mol%). This transformability value hints that t2 in principle is still in the transformable range and not yet t'-phase (starting at <1.012) [40]. The domain size is however small (<32 nm calculated from peak width) and it is extremely unlikely that such a small and highly stabilized domain located in the inside of a grain can actually be transformed [41,42].

For the composition at 1300 °C we would then either end up with $V_{trans} = 0.8$ considering only the cubic or with $V_{trans} = 0.5$ by subtracting the volume content of t2 phase. The consequence is that h= 1.1 µm for $V_{trans} = 1$, h =1.5 µm for $V_{trans} = 0.8$ or h = 4.5 µm for $V_{trans} = 0.5$. As the transformation toughness scales with $V_f \cdot \sqrt{h}$ [30] (E = Young's modulus, $\nu$ = Poisson's ratio = 0.31, volume change upon transformation $\varepsilon_T = 0.05$ and X = transformation efficiency; X = 0.22 for pure dilatation and X = 0.48 for dilatation and shear) the calculated transformation toughness may change by a factor of 2 by only considering different assumptions for $V_{trans}$. Assuming dilatation and shear (X = 0.48) transformation toughness contributions may thus account for $\Delta K_{IC}^T = 3.75$ MPa$\sqrt{}$m for $V_{trans} = 0.8$ or 6.5 MPa$\sqrt{}$m for $V_{trans} = 0.5$. Different assumptions for X leads to further uncertainties.

$$\Delta K_{IC}^T = X\cdot E/(1 - \nu)\cdot V_f\cdot \sqrt{h}\cdot\varepsilon_T. \qquad (2)$$

A plot of fracture resistance vs. $V_f \cdot \sqrt{h}$ (h calculated for $V_{trans} = 0.8$) shows that if we compare "coated" Y-TZP materials sintered at different temperatures, there is no linear correlation (Figure 10) as it was reported for co-precipitated materials [43].

These "coated" Y-TZPs temperatures in spite of their identical net composition are completely different materials depending on which sintering conditions are applied. Based on this set of data it is impossible to determine $K_{tip}$, the intrinsic toughness by extrapolating to $V_f\cdot\sqrt{h} = 0$. Assuming $K_{tip} = 4$ MPa$\sqrt{}$m as quoted by Swain [43] for co-precipitated Y-TZP we may well explain the data points for the materials sintered at 1400–1450 °C but not the materials sintered at lower temperature. In an earlier publication on Y-TZP sintered at different temperatures [24] the threshold toughness $K_{IO}$ (not identical but closely correlated to $K_{tip}$) was measured and the R-curve dependent part of toughness was determined in a SIGB (stable indentation crack growth in bending) measurement. $K_{IO}$ = 3.8 MPa$\sqrt{}$m between 1350–1450 °C and increases linearly to 5.5 MPa$\sqrt{}$m for a sintering temperature of 1300 °C. The R-curve dependent toughness falls from 5.5–6 MPa$\sqrt{}$m at 1300–1350 °C to 2 MPa$\sqrt{}$m at 1450 °C. The data collected in this study and the results of previous studies on 3Y-TZP and 3Yb-TZP made from identical starting powders fit quite well into this explanation model [23,24].

Based on experimental evidence (and not just a priori as in [23]) we may now suggest that not only the slope of the curve and thus the transformation mechanism changes from dilatation and shear for low sintering temperature to predominantly dilatoric for high temperature but that also the intrinsic toughness is different for different sintering temperatures.

It is assumed that the transformation characteristics change continuously with temperature due to the proceeding yttria diffusion from Mg-PSZ like behavior (X = 0.48) at low sintering temperature to a classical Y-TZP behavior (X = 0.27) at high sintering temperature.

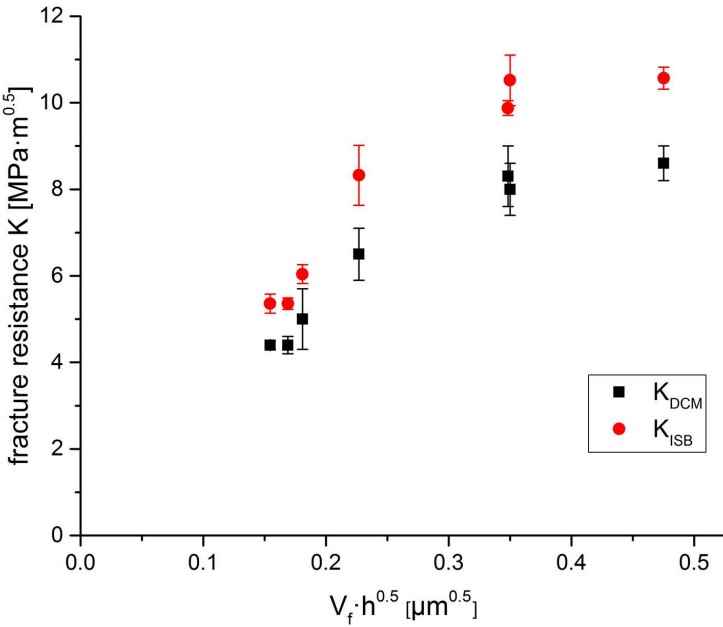

**Figure 10.** Fracture resistance vs. Vf$\sqrt{}$h (h calculated assuming $V_{trans}$ = 0.8).

## 5. Summary and Conclusions

XRD measurements have shown that the transition in mechanical properties, microstructure and ageing characteristics of 3Y-TZP manufactured from stabilizer coated powder can be traced back to a change in the phase composition of the materials with increasing time and temperature. Two extreme cases may be considered:

Case A: Y-TZP sintered at low temperature is extremely fine grained, has a purely intergranular fracture behavior, high fracture resistance and LTD resistance. The phase analysis shows that the material consists of grains with a core-shell structure. While the core has a low yttria content and high tetragonality the cubic shell has a very high yttria content.

Case B: material sintered at high temperature has a coarser microstructure the tetragonal phase has the maximum possible yttria content of 2.5 mol%, a lower tetragonality and the remaining cubic phase has an yttria content of only ~4.5 mol%. The fracture and LTD are "regular" i.e., similar to co-precipitated 3Y-TZP.

With increasing sintering temperature, the composition and properties gradually shift from case A to case B due to progressive diffusion of yttria.

The predominant influence of the diffusion phenomena leads to the surprising fact that in case of co-precipitated 3Y-TZP grain size and fracture resistance are coupled while they are de-coupled for Y-TZP made from stabilizer coated powder sintered at low temperature. It is moreover quite probable that not only the transformability of the zirconia but also its transformation efficiency changes with progressive yttria diffusion.

Implications to the typical target applications of 3Y-TZP are evident. The state-of-the-art Y-TZP materials only become tough in extremely overfired state where grain sizes are large resulting in very poor LTD resistance. Y-TZP materials made from coated powders can combine high toughness and damage tolerance with high ageing resistance.

In most applications, pressure-assisted sintering as in this study is not be applicable but components shaped by pressing, casting and injection molding are pressurelessly sintered in air. The challenge in manufacturing of components is now to develop materials which have a very high sinterability at low temperature in order to obtain a fully dense material before the stabilizer redistribution is finished and the favorable properties are lost. This represents a challenge in the field of powder selection, powder technology, shaping and sintering.

**Supplementary Materials:** The following are available online at http://www.mdpi.com/2571-6131/2/2/22/s1, the supplementary materials contain additional original SEM images of polished and thermally etched surfaces as well as of fracture surfaces for all applied sintering temperatures.

**Funding:** This research received no external funding.

**Acknowledgments:** Diego Elmer and Felicitas Predel (MPI-FKF) are gratefully acknowledgement for sample preparation and SEM.

**Conflicts of Interest:** The authors declare no conflict of interest.

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
