# Peer review of "Evidence of Phase Transitions and Their Role in the Transient Behavior of Mechanical Properties and Low Temperature Degradation of 3Y-TZP Made from Stabilizer-Coated Powder"

_ceramics, doi:10.3390/ceramics2020022_

Round 1

Reviewer 1 Report

This is a very interesting study which provides some useful information for further investigations. Only minor comments are suggested to improve the quality of this manuscript, as addressed below:

Line 302-303, the authors describe “The transition between these two extremes is reached at a sintering temperature of 1375°C here coexistence of two tetragonal and two cubic phases is observed”. However, only 3 different crystal phases were shown at the rightmost of stage I in figure 9.

Line 291-293, the authors also describe “… vanish at 1425°C, the high yttria cubic phase c1 … disappears at the same temperature”. In figure 9, the diagram of 1375°C doesn’t show the c1 phase.

I believe that figure 9 is the highlight of this manuscript. It is suggested that the authors check all the text in the manuscript to fit the model of figure 9.

Author Response

Line 302-303, the authors describe “The transition between these two extremes is reached at a sintering temperature of 1375°C here coexistence of two tetragonal and two cubic phases is observed”. However, only 3 different crystal phases were shown at the rightmost of stage I in figure 9.

Line 291-293, the authors also describe “… vanish at 1425°C, the high yttria cubic phase c1 … disappears at the same temperature”. In figure 9, the diagram of 1375°C doesn’t show the c1 phase.

both comments: The figure 9 was modified so that description in text and the measurements (fig. 6) match the schematic description given in  figure 9.

Reviewer 2 Report

The paper is acceptable for publication after minor revision in the English language (terms and syntax).

This work reports interesting contribution to the understanding of the phase transformation that occurs during sintering of 3YSZ produced by a “coating” process.

The aim of this paper is to discuss what happens during sintering of 3YSZ produced by “coating” with respect to the diffusion of yttria and phase transformation, showing the sequence of the events and the temperatures in which they occur, contributing to the understanding of the sintering process to obtain a material suitable for use with the appropriate mechanical properties.

The methods used are well described and the data obtained are presented and discussed clearly.

The work presents an important contribution to the area since it analyzes the mechanical and structural properties of 3YSZ for different sintering temperatures, associating the crystalline structure and yttria diffusion by the grain and grain boundary during the sintering process.

Author Response

no response to requests required

Reviewer 3 Report

1. How did the author estimate the fraction of transgranular fracture (~30 %) in the fractured surface of 3Y-TZP hot-pressed at 1450°C/1h? If the author refer to the numerical value, it might be fair to mention how the author estimated the fraction in the manuscript.

2. How much was the fraction of transgranular fracture in the fracture surface of the other samples (1300°C/1h and 1375°C/1h)? If the author refer to the numerical value of the fraction of transgranular fracture, it might be fair to mention the numerical value for the other samples.

3. This reviewer did not feel it was very fair to conclude that the fraction of transgranular fracture distinctly increased with increasing sintering temperature from Figs. 1(d)~(f), because the number of grains in the observation area was very different. According to this reviewer, the explanation about Figs. 1(d)~(f) might be more persuasive if there are also the SEM images of 3Y-TZP hot-pressed at 1300°C/1h, 1375°C/1h, and 1450°C/1h in which almost the same number of grains are present.

4. Was the composition of each grain almost the same? The grain size distribution was broad. There might be compositional difference between comparatively large and small particles. Did the author investigate the composition of each grain by TEM or EDX? If yes, those results should be added in the manuscript or the supplementary information.

5. The value of the vertical axis in Fig.6 should not be presented.

6. This reviewer could not understand the discussion about the stage II in Fig. 9 well. If the formation and disappearance of yttria-rich sub-grain-boundary occur, then the diffusion of yttria from the whole part of the shell to the bulk should also occur in the same time scale (then the core-shell structure should disappear in the same time scale). Could the author better explain why only the part of the shell contacted with another grain disappears while the other part of the shell remains?

7. The author claims that the former sub-grain boundaries are visible as lines with slightly darker shade in large grains in Fig. 1(c). If such small compositional difference between the bulk and sub-grain boundaries could be observed by SEM, then the core-shell structure of each grain should be visible in Figs.1 (a) and (b). Could the author explain why the shell of each grain was invisible in Figs.1 (a) and (b) while sub-grain boundaries were visible in Fig.1 (c)? Please also add the information of compositional analysis of the sub-grain boundaries.

Author Response

How did the author estimate the fraction of transgranular fracture (~30 %) in the fractured surface of 3Y-TZP hot-pressed at 1450°C/1h? If the author refer to the numerical value, it might be fair to mention how the author estimated the fraction in the manuscript.

The given values were approximated from SEM images of different resolution.

2. How much was the fraction of transgranular fracture in the fracture surface of the other samples (1300°C/1h and 1375°C/1h)? If the author refer to the numerical value of the fraction of transgranular fracture, it might be fair to mention the numerical value for the other samples.

for 1300°C the text passage is unambigous and says "the crack mode is purely intergranular", this means 100%.

Some changes were made in the text to indicate that the values are rough estimations

3. This reviewer did not feel it was very fair to conclude that the fraction of transgranular fracture distinctly increased with increasing sintering temperature from Figs. 1(d)~(f), because the number of grains in the observation area was very different. According to this reviewer, the explanation about Figs. 1(d)~(f) might be more persuasive if there are also the SEM images of 3Y-TZP hot-pressed at 1300°C/1h, 1375°C/1h, and 1450°C/1h in which almost the same number of grains are present.

An extensive addition of SEM images to the manuscript as such. To give the reader a better impression additional SEM images will be provided in the supplement.

4. Was the composition of each grain almost the same? The grain size distribution was broad. There might be compositional difference between comparatively large and small particles. Did the author investigate the composition of each grain by TEM or EDX? If yes, those results should be added in the manuscript or the supplementary information.

Unfortunately TEM and EDX measurements were not made in this study and cannot be provided. In earlier studies on YbNd-TZP YNd-TZP and GdNd-TZP EDX measurements confiremed that the larger grains contain much more stabilizer than the small ones. This is not surprising and in line with literature as in cubic grains there is no solute drag and no grain growth inhibition.

5. The value of the vertical axis in Fig.6 should not be presented.

was changed

6. This reviewer could not understand the discussion about the stage II in Fig. 9 well. If the formation and disappearance of yttria-rich sub-grain-boundary occur, then the diffusion of yttria from the whole part of the shell to the bulk should also occur in the same time scale (then the core-shell structure should disappear in the same time scale). Could the author better explain why only the part of the shell contacted with another grain disappears while the other part of the shell remains?

The problem is caused by an inaccuracy in the figure 9 which was now corrected.

The bulk is saturated with yttria up to the level of the t/t+c boundary and cannot take up any more yttria without changing its phase composition. The surplus yttria dragged inside the merged grain must diffuse OUT of the bulk to the boundary (formation of a cubic nucleus inside the grain is unlikely). The surplus yttria in the shell for the same reason cannot diffuse into the grain as this is thermodynamically forbidden (the t+c field is a miscibility gap). Threfore initially when grain growth happens the yttria concentration in the shell will actually rise as the the only drain for yttria accumulated in the shell is phase separation and formation or growth of cubic grains.

7. The author claims that the former sub-grain boundaries are visible as lines with slightly darker shade in large grains in Fig. 1(c). If such small compositional difference between the bulk and sub-grain boundaries could be observed by SEM, then the core-shell structure of each grain should be visible in Figs.1 (a) and (b). Could the author explain why the shell of each grain was invisible in Figs.1 (a) and (b) while sub-grain boundaries were visible in Fig.1 (c)? Please also add the information of compositional analysis of the sub-grain boundaries.

With respect to the SEM images it should be considered that the surfaces were thermally etched. the etching reveals the grain boundaries and leads to a certain uplift of the boundary region which conincided with the postulated core-shell boundary. The etching however also reveals in some larger grains merged from smaller units the "halo" of the former grain boundaries of the smaller grains. Of course The SEM is not able to measure stabilizer concentrations.

Reviewer 4 Report

dear Author,

the MS is of interest for the readers and generally well written.

However, I have some comments aimed to improve your paper before publishing.

I think it is important to include the SEM images of the samples sintered at 1400°C since this temperature represents more than (or in addition to) 1375°C a discontinuity point for most of the material properties (see Fig.2-3-4-5-(6)-7-8 and discussion fig 9)

Give an evidence (just include an image inset) of the nanoscale roughness cited at row 166 and of the alumina grains (EDS, BSE?).

Row 215 and 217: tetragonality seems to me 1.023 (fig7)

Row 233: typos error Aa

Row 243-245 looking at fig.8 it seems to me that Vm are identical after 3h of aging at low T and about 4-6% while at high T is about 10%.

Row 250: Change 'stay practically unaffected' with 'are slightly affected' (is <15%)

Row 251:not true if we take into account the bar errors...

Row 294: in figure 9 is reported t2 contains 1mol% and t1 2,5 mol% of Y2O3

Row 302: typos error 'here'

Row 431: typos error 'as in used'

Row 437: typos error: 'acknowledgement'

Author Response

the MS is of interest for the readers and generally well written.

However, I have some comments aimed to improve your paper before publishing.

I think it is important to include the SEM images of the samples sintered at 1400°C since this temperature represents more than (or in addition to) 1375°C a discontinuity point for most of the material properties (see Fig.2-3-4-5-(6)-7-8 and discussion fig 9)

The supplement contains SEM images (magnification 20000x) of thermally etched microstructures and fracture surfaces for all sintering temperatures.

Give an evidence (just include an image inset) of the nanoscale roughness cited at row 166 and of the alumina grains (EDS, BSE?).

EDS/BSE images were not made and are not readily available. A characteristic image wasadded to the supplement

Row 215 and 217: tetragonality seems to me 1.023 (fig7)

no value is correct !

Row 233: typos error Aa

done "As..."

Row 243-245 looking at fig.8 it seems to me that Vm are identical after 3h of aging at low T and about 4-6% while at high T is about 10%.

The description in this section was changed to absolute values instead of added increments to make this more clear

Row 250: Change 'stay practically unaffected' with 'are slightly affected' (is <15%)

done

Row 251:not true if we take into account the bar errors...

was changed

Row 294: in figure 9 is reported t2 contains 1mol% and t1 2,5 mol% of Y2O3

figure 9 was changed

Row 302: typos error 'here'

done

Row 431: typos error 'as in used'

done

Row 437: typos error: 'acknowledgement'

done

Round 2

Reviewer 3 Report

I recommend that this paper be accepted.

Reviewer 4 Report

The revised manuscript is acceptable in its form.